# Synthesis, Self-Assembly in Crystalline Phase and Anti-Tumor Activity of 2-(2-/4-Hydroxybenzylidene)thiazolo[3,2-*a*]pyrimidines

**DOI:** 10.3390/molecules27227747

**Published:** 2022-11-10

**Authors:** Artem S. Agarkov, Anna A. Nefedova, Elina R. Gabitova, Alexander S. Ovsyannikov, Syumbelya K. Amerhanova, Anna P. Lyubina, Alexandra D. Voloshina, Pavel V. Dorovatovskii, Igor A. Litvinov, Svetlana E. Solovieva, Igor S. Antipin

**Affiliations:** 1Arbuzov Institute of Organic and Physical Chemistry, FRC Kazan Scientific Center, Russian Academy of Sciences, Arbuzova 8, 420088 Kazan, Russia; 2Department of Organic and Medical Chemistry, Kazan Federal University, Kremlevskaya 18, 420008 Kazan, Russia; 3National Research Center “Kurchatov Institute”, Akademika Kurchatova 1, 123182 Moscow, Russia

**Keywords:** thiazolo[3,2-*a*]pyrimidines, solid-state structure, non-covalent interactions, cytotoxicity, antitumor agents, multiplex analysis of DNA markers of genotoxicity

## Abstract

A series of new thiazolo[3,2-*a*]pyrimidines different by aryl substituents in 2 and 5 positions are synthesized and characterized in solution as well as in the crystalline phase using ^1^H and ^13^C NMR-, IR-spectroscopies, mass-spectrometry methods, and single crystal X-ray diffraction (SCXRD). The SCXRD study revealed the role of intermolecular H-bonding in the formation of supramolecular architectures (racemic monomers, centrosymmetric racematic dimers, or homochiral 1D chains) of obtained thiazolo[3,2-*a*]pyrimidines derivatives depending on solvents (aprotic DMSO or protic EtOH) used upon the crystallization process. Moreover, the in vitro study of cytotoxicity toward different tumor cells showed their high or moderate efficiency with moderate cytotoxicity against normal liver cells which allows to consider the obtained thiazolo[3,2-*a*]pyrimidine derivatives as promising candidates for application as antitumor agents.

## 1. Introduction

Heterocycles, especially those composed of different heterocyclic aromatic rings, generally possessing high biological activity represent organic compounds widely used for design of new antitumor drugs [1,2]. Among the various heterocyclic compounds in terms of antitumor activity, the thiazoles and pyrimidines derivatives are some of the most attractive precursors in medical chemistry due to their high pharmacological application in particular as anti-tuberculosis drugs, antioxidants, antimicrobial, antiviral agents, as medicaments for inflammatory disease treatment and antitumor therapy [3,4,5,6,7,8]. Nowadays, it is established that cancer of various types is a leading cause of morbidity and mortality worldwide, characterized by irregular cell growth. The number of deaths caused by cancer is still growing. According to the World Health Organization (WHO), in 2020, cancer accounts for more than 19 million new cases registered and ranks as one of the first leading cause of death in people under 70 years old [9]. For this purpose, in the past decades, the design of new heterocyclic compounds that demonstrate high antitumor activity both with low and normal cell cytotoxicity still attracts huge attention.

Recently, it was shown that due to high biological and pharmacological activities, heterocyclic compounds based on functional derivatives of 2-arylmethylidene thiazolo[3,2-*a*]pyrimidines are appealing molecules demonstrating a high potential in anticancer drug design. In particular, it was demonstrated that indole derivative of triazolopyrimidine **1** (Figure 1) exceeded the cytotoxicity efficiency of the widely used antitumor drug doxorubicin by 1.15 times in the MCF-7 breast cancer cell line [10].

In addition, compound **1** has demonstrated high absorption ability in the gastrointestinal tract due to optimal hydrophilicity/lipophilicity of the molecule [11]. The related compound **2** (Figure 1) has been used for the growth of Bcl-2 proteins inhibitor which leads to many types of tumor cells apoptosis [12]. Compound **3** (Figure 1) has displayed remarkable inhibiting activity of the anti-apoptotic Bcl protein with IC_50_ value of 3.4 μM, which practically corresponds to the one established for *obatoclax* drug [13]. Other similar compounds **4**–**6** containing thiazolo[3,2-*a*]pyrimidine fragment were found to be promising, and selective CDC25 phosphatase or casein kinase inhibitors play a key role in the regulation of the cell cycle observed in many types of cancer [14,15]. Earlier it was also reported that the triazolpyrimidine derivatives containing another molecular junction at C2 atom (dihydrazone derivatives) showed low cytotoxicity against normal and tumor cell lines [16].

Earlier, we reported on the synthesis and crystal structures of three derivatives of 2-(2-hydroxybenzylidene)thiazolo[3,2-*a*]pyrimidines containing phenyl-(**7**), 2-hydroxyphenyl-(**8**) and 4-methoxyphenyl-(**9**) substituents at the C5 carbon atom, where the conglomerate crystallization of one of the studied heterocycles (compound **8**) based on thiazolo[3,2-*a*]pyrimidine (*P*2_1_2_1_2_1_ space group) into chiral 1D chains stabilized by intermolecular H-bonding is presented [17]. It should be mentioned that up to now no anti-tumor activity of the described compounds is studied.

Following this line, we synthesized three new compounds, **10**–**12,** bearing 2-hydroxybenzylidene substituent at the C2 carbon atom. In addition, in order to expand this series of new thiazolo[3,2-*a*]pyrimidines, the 4-hydroxybenzylidene derivatives were obtained: compound **13** (isomeric to compound **7**), compound **14** (isomeric to compound **12**), and compound **15** (isomeric to compound **10**) (Figure 2). Herein the crystal structure of the obtained compounds **9**–**14** is discussed below. Moreover, the study of anti-tumor activity of a series of obtained 2-and 4-hydroxybenzylidene thiazolo[3,2-*a*]pyrimidine derivatives **7**–**15** is also presented.

## 2. Results and Discussion

All arylmethylidenethiazolopyrimidines **7**–**15** were synthesized following the scheme presented on Figure 2. In the first stage, a three-component Biginelli condensation involving appropriate aldehyde (benzaldehyde, 2-methoxybenzaldehyde, 4-methoxybenzaldehyde, 3-nitrobenzaldehyde, or 4-brombenzaldehyde), thiourea, and 1,3-dicarbonyl compound (acetoacetic ether or benzoylacetone) in a molar ratio 1:1.5:1 led to the preparation of 1,2,3,4-tetrahydropyrimidine-2-thions in the presence of a catalytic amount of molecular iodine (0.03 mmol) under refluxing conditions in acetonitrile as described earlier [18,19,20,21,22]. Then the obtained compounds were used as precursors for the synthesis of targeted 2-and 4-hydroxybenzylidene thiazolo[3,2-*a*]pyrimidine **7**–**15** by reaction with corresponding 2- or 4-hydroxybenzaldehydes. The desired products were obtained by filtration of the reaction mixture followed by the recrystallisation from the methanol or ethanol solution in high yields.

The structures of synthesized compounds were studied using SCXRD, whose data are in agreement with the structure of these compounds in solution (for ^1^H-,^13^C-NMR-, IR-, and mass-spectra see Appendix A). In the crystalline phase, all the obtained compounds present similar structure when all carbon atoms, except C5, as well as hetero atoms (S and N) of bicyclic thiazolo[3,2-*a*]pyrimidines fragment are located in the same plane. The asymmetric *sp^3^* hybridized C5 atom is found to be disposed out of thiazolo[3,2-*a*]pyrimidine plane displaying the distance of 0.126–0.273(1) Å and leading to a *sofa* conformation of six-membered pyrimidine ring (see Appendix A). Interestingly, it was found out that for **12**, C5 atom is found to be located in the same plane as thiazolo[3,2-*a*]pyrimidine bicyclic ring. All obtained molecules display only one C=C double bond configuration when 2- or 4-hydroxybenzylidene fragment and S atom of thiazolyl moiety are *cis*-orientated (*Z*-isomers).

The aryl substituent connected to C5 atom occupies an axial position and is disposed in the plane bisecting N4-C5-C6 angle. For **11**, the intramolecular O–π bonding is observed between the *ortho* OCH_3_ group of phenyl substituent disposed at C5 atom and the π-system of pyrimidine moiety (d_O17-C3N2centroid_ = 2.731 Å). Such type of interaction leading to stabilization of molecular conformation have been already reported recently [17,23].

Both the distances observed for C2–C9 and C9–C10 bonds as well as the dihedral angle formed between the phenol moiety of benzylidene substituent at C2 atom and thiazolo[3,2-a]pyrimidine bicyclic ring evidence the formation of a large electron-conjugated system for all compounds (see Appendix A). In the crystalline phase, all studied 2-hydroxybenzylidene)thiazolo[3,2-a]pyrimidines adopt conformation when 2-OH group is found to be syn orientated with carbonyl group of thiazolyl moiety.

For ester derivatives **8**–**10** and **12**–**14**, the carbonyl group positioned at C6 atom is found to be conjugated with thiazolo[3,2-a]pyrimidine system which was attested by the values of the corresponding dihedral angle (see Appendix A). In contrast, the ketoaryl derivative **11** displays relatively large dihedral angle with thiazolo[3,2-a]pyrimidine fragment equal to 39°. Other characteristic bond distances and angles gathered in the Appendix A are found to be similar for all the studied compounds.

Due to presence of H-donor such as and H-acceptor atoms within the molecular structures, one may assume that the obtained thiazolo[3,2-a]pyrimidines are prone to be involved in supramolecular assembles formation in the crystalline phase as well as in solution. It is worth noting that the propensity of such type of molecules to participate in spontaneous self-assembly upon the H-bonding may play the important role in their medical application [24,25]. Indeed, as it was established, the supramolecular organization of obtained compounds in the crystalline phase depends on the mutual orientation of H-donor/acceptor in the molecular scaffold as well as on the use of aprotic or protic solvent.

For 2-hydroxybenzylidene derivatives **9**, **11,** and **12**, when crystallized from the ethanol solution, the formation of supramolecular racemic dimers displaying the centrosymmetric structure resulting from the intermolecular H-bonding between O11 atom of hydroxyl group and O3 atom belonging to carbonyl group of thiazolyl ring with the O…O distance range of 2.612(2)–2.797(2) Å was observed (see Figure 3, Appendix A).

Whereas the self-assembly of dimers based on **9** led to the formation of 2D supramolecular layer due to π-staking of thiazolo[3,2-a]pyrimidines moieties [17], and for **11**, it was found out that ketophenyl C6 atom substituent displays significant π-bonding (d_C6centroid/C6centroid_ = 3.660 Å, dihedral angle equal to 0°) producing the chains (see Figure 4a). In addition, the obtained chains through interacting with each other through intermolecular pyrimidine-(2-hydroxy)phenyl π-bonding (d_C3N2centroid/C6centroid_ = 3.670 Å, dihedral angle equal to 0°) also results in 2D supramolecular layers running along (022) crystallographic plane.

The changing of 2-or 4-metoxyphenyl substituent at C6 atom in **11** and **9** by 4-bromophenyl fragment in **12** surprisingly led to the formation of 1D chain composed of H-bonded supramolecular dimers stacked through CH/π-bonding of C23 atom of ester group with bromophenyl moiety of adjacent molecules of **12** (d_C23-C6centroid_ = 3.505 Å) (see Figure 4b).

In crystal, all supramolecular dimers **9**, **11,** and **12** demonstrate the parallel crystal packing. No solvent molecules were found in the unit cell of **9**, **11,** and **12**.

Compounds **9** and **12** were also crystallized from DMSO solution. The SCXRD revealed the destroying of the H-bonded dimer assembly caused by the formation of solvate complexes **9-DMSO** [17] and **12-DMSO** in which solvent molecule was H-bonded via O30 atom with O11 atom of hydroxyl group of heterocyle (see Appendix A, Figure 5). This is a remarkable example of the control on the self-assembly process using aprotic solvent displaying high H-accepting ability. In crystal of **12-DMSO**, the solvate complexes are stacked into the chains due to weak π–π and Br/π bonding (see Appendix A). It should be noted that in **9-DMSO** and **12-DMSO,** molecules **9** and **12** adopt syn-orientation of carbonyl and hydroxyl groups and present flattened structure which has been already observed in the case of supramolecular H-bonded dimers **9**, **11,** and **12** (see Figure 3).

The use of methanol as a solvent upon the crystallization of compound **10** led to the generation of another solvate complex **10-MeOH** presenting the formation of zigzag homochiral chains. The MeOH molecules in **10-MeOH** forming the H-bonds with N8 atom of pyrimidine fragment and O11 atom of 2-hydroxybenzylidene moiety act as bridges connecting two adjacent thiazolo[3,2-a]pyrimidine molecules (see Appendix A, Figure 6a). Within the chains, heterocyclic molecules display coplanar orientation. A similar connectivity pattern has been already observed earlier for **7-EtOH** [17], but **10-MeOH** displays a OON angle between the hydrogen bonds equal to 106.37(5)°, which is less compared to the one observed for **7-EtOH** (157.90°). In crystal, the 1D chains are stacked parallelly forming 2D layers along (002) crystallographic plane due to π–π interaction between the thiazolyl groups (d_C2-C3NScentroid_ = 3.494 Å, dihedral angle equal to 0°) and 2-hydroxybenzylidene moieties (d_C9-C6centroid_ = 3.561 Å, dihedral angle equal to 0°) (Figure 6b).

In order to study the influence of hydroxyl group position within the structure of benzylidene substituent, the crystals of new 4-hydroxybenzylidenethiazolo[3,2-a]pyrimidines **13** and **14** were obtained by slow evaporation of their EtOH or DMSO solutions and analysis using SCXRD (see Experimental part). It was established that a self-assembly of **13** and **14** in the absence of strong H-acceptors results in 1D infinite zigzag homochiral chains formation involving the intermolecular H-bonding between O13 atom of 4-hydroxyl group acting as H-donor with.N8 atom of pyrimidine fragment behaving as H-acceptor (see Appendix A, Figure 7). For **13**, a weak chalcogen bonding between the O atom from hydroxyl group and S-atom from thiazolyl ring is also observed (d_O13A-S1B_ = 3.301(6) Å).

Although **13** and **14** demonstrate the similar supramolecular motif of zigzag chain and dihedral angle between adjacent heterocyclic molecules within the chain structure equal to 65°, they are different by their crystal packing. In crystal of **13**, the obtained chains are stacked without any specific intermolecular interactions. For **14**, in contrast to **13**, the formation of 3D supramolecular network is observed resulting from the π-stacking thiazolyl and 4-hydroxybenzylidene moieties of heterocyclic molecules belonging to neighboring chains (d_C10-C3NS_ = 3.566 Å, dihedral angle equal to 0˚) as well as the intermolecular O–π bonding involving O22 atom and thiazolyl moieties (d_O22-C3NScentroid_ = 3.098 Å, φ = 142.27˚) (see Figure 8 and Appendix A).

As for **9**, **11,** and **12**, the use of DMSO as a solvent upon crystallization afforded the formation of solvate **14-DMSO** in the crystalline phase where DMSO was found to be H-bonded with O11 atom belonging to hydroxyl group of **14**. It is worth noting that involvement of DMSO in H-boding with heterocyclic compound led again to the destruction of the infinite polymeric chains (Figure 9) observed for **13** and **14** when crystallized from ethanol solution. As for **12-DMSO**, **14-DMSO** molecules are stacked in the crystalline phase into the chains resulting from the halogen–π bonding displaying a shorter distance between Br atom and thiazolyl group equal to d_Br1-C2_ = 3.226(1) Å (see Appendix A) with respect to **12-DMSO** (see Appendix A).

The most important stage in the investigation of new chemical compounds able to be used as potential drugs is the study of their cytotoxic properties. All prepared compounds were tested for their cytotoxicity against cancer and normal cell lines (Table 1). Data on cytotoxic activity are represented by IC_50_ values (the concentration which causes the death of 50% of cells in the experimental population).

The tested compounds showed high and moderate activity against a number of cancer lines of various genesis and demonstrated moderate cytotoxicity against normal liver cells. The most significant results were shown in the case of cell lines of cervical carcinoma (M-HeLa) and human duodenal adenocarcinoma (HuTu80). Their cytotoxic effect was manifested at the level of the comparison drug sorafenib, and for **7**, with respect to M-HeLa, it was found to be doubly increased compared to the known antitumor drug.

The selectivity of compounds against cancer cells is an important criterion for assessing the cytotoxic effect. For this purpose, the selectivity index (SI) was calculated as a ratio between the IC_50_ value for normal cells and the IC_50_ value for cancer cells. The selectivity index values for the tested compounds are shown in Table 2. It can be seen that the highest selectivity against the cancer lines M-HeLa and HuTu 80 was demonstrated by compound **7**, whose SI value was 6.0 and 7.5, respectively. Generally, the compounds with SI ≥ 3 [26] are considered selective. According to these data, compound **7** can be considered selective toward the M-HeLa and HuTu 80 cell lines. It was worth noting that the reference drug sorafenib showed significantly lower selectivity with respect to **7**.

Apoptosis is one of the most important mechanisms used for screening new anticancer agents. Apoptotic effects in M-HeLa cells were evaluated by flow cytofluorometry using annexin V kits that can bind phosphatidylethylserine. Phosphatidylserine is contained in minimal amounts on the membrane surface of healthy cells. Therefore, the interaction of annexin V with these cells is insignificant. During apoptosis, phosphatidylserine molecules can interact with the protein. This interaction leads to an increase in the fluorescence intensity of apoptotic cells which is detected by a flow cytofluorimeter. Apoptosis-inducing effect was evaluated using the leading compound **7** at concentrations of IC_50_/2 and IC_50_ on the M-HeLa cell line (see Figure 10). After 24-h incubation in the presence of **7**, dose-dependent apoptosis was observed in M-HeLa cells. Moreover, the observed effects were more active at the late stage of apoptosis.

Evaluation of the mitochondria functions of the cell is another universal sign of apoptosis characteristic of eukaryotes [27]. With mitochondrial dysfunction, proapoptotic factors are released into the cytoplasm—cytochrome C, AIF, Smac/DIABLO, endonuclease G, as well as proforma caspases 2, 3, and 9—inducing cascade of caspases [28]. The output of these protein factors can be realized both by breaking mitochondrial membranes and by activating specific channels in the outer membrane of mitochondria. The events described above are usually accompanied by a change in the permeability of the inner membrane of mitochondria for H^+^ protons, which leads to a change in the membrane potential of mitochondria (ΔΨm). Methods for studying the membrane potential of mitochondria using flow cytofluorimetry are based on the use of cationic lipophilic dyes, which in the literature are called “mitochondrial probes”. The principle of operation of these dyes is very simple—lipophilic probes spontaneously penetrate bilipid membranes (the surface membrane of the cell, as well as the outer and inner membranes of mitochondria) and accumulate in areas with a high pH value, that is, under the inner membrane of mitochondria. This effect results in a change of the intensity of cell fluorescence observed during the analysis on a flow cytofluorimeter. In this study, the fluorescent dye JC-10 from the Mitochondria Membrane Potential Kit was used to assess the change in ΔΨm under the action of compound **7** at concentrations of IC_50_/2 and IC_50_ on the M-HeLa cell line. JC-10 accumulates in the mitochondrial matrix and forms aggregates (J-aggregates) with red fluorescence in normal cells evidencing a high potential of the mitochondrial membrane. The mitochondrial membrane potential was found reduced in apoptotic cells which can be related with JC-10 starting to diffuse from the mitochondria and turning into a monomeric form (J-monomer) exhibiting green fluorescence. A decrease in the mitochondrial membrane potential of M-HeLa cells was observed after 24 h of treatment with the leading compound **7**. The effect becomes more significant with an increase in the concentrations of the tested compounds up to IC_50_ (see Figure 11 and Figure 12). The obtained results confirm that the mechanism of cytotoxic action of **7** can be associated with the induction of apoptosis along the internal mitochondrial pathway.

An increase in the production of reactive oxygen species (ROS) by compounds also characterizes the development of apoptosis along the mitochondrial pathway. Mitochondria are both a potential source and target of ROS. An increase in ROS production leads to the disruption of mitochondrial functions and, subsequently, to irreversible damage to cells. In this regard, the effect of the leader compound **7** at IC_50_/2 and IC_50_ concentrations on ROS production in M-HeLa cells was investigated using flow cytometry analysis and CellROX^®^ Deep Red flow cytometry kit. The data presented in Figure 13 show a significant increase in the fluorescence intensity of CellROX^®^ Deep Red compared to the control (unpainted cells). This indicates an increase in ROS production in the presence of the tested compound **7**.

The effect of cytotoxic agents in general may be associated with a violation of the passage of eukaryotic cells phases of the cell cycle. This synchronizes and slows down the proliferation of a population of rapidly multiplying cells. The analysis of the cell cycle by quantifying the DNA content in the cell is a reliable research method that allows us to assess at which phase the cell cycle was stopped. This method allows us to study the phases of the cell cycle: the distribution of cells by G1/G0-, S-, and G2/M-phases of the cell cycle is estimated by determining the relative DNA content in cells using DNA-binding fluorescent dyes such as PI (propidium iodide), 7-AAD (7-aminoactinomycin), DAPI (4′6′-diamidino-2-phenylindole (DAPI)), Hoechst, SybrGreen, etc. Anomalies of the cell cycle detected on the histogram of DNA content frequencies are often observed after various types of cell damage, especially after exposing to chemical agents. In this work, cell cycle studies were carried out using the example of the leader compound **7** at concentrations of IC_50_/2 and IC_50_ on the M-HeLa cell line using the fluorescent dye propidium iodide (PI). The fluorescence intensity of the stain cells correlated with the amount of DNA contained in them. The results of the cell cycle analysis after treatment with compound **7** for 24 h at concentrations of IC_50_/2 and IC_50_ on the M-HeLa cell line showed a significant delay of cells in the G0/G1 phase compared with the reference sample(Figure 14).

The processes of DNA damage and repair constantly occur in living cells. They can be caused by natural causes as a result of cellular respiration and metabolism or exposure to various chemical agents. Restoration of DNA damage is necessary for the normal functioning of cells and their preservation in a healthy state. Many proteins are involved in the processes of detecting and repairing DNA damage, among which groups of sensor proteins, mediators, converters, and effectors should be noted. Sensory proteins, such as Rad9, Rad1, and Hus1, accumulate at the site of DNA damage and provide phosphorylation of checkpoint proteins, which is influenced by intracellular kinases—ATM (ataxia-telangiectasia mutated) and ATR (ataxia-telangiectasia). Activation of mediator proteins such as H2AX, BRCA1, and SMC1, leads to stable protein–protein interaction, which facilitate the transmission of signals to intracellular kinases and at the same time activate checkpoint kinases. The kinases of the control points Chk1 and Chk2 are necessary to stop the cell cycle of the main control points, G1/S and G2/M, on which the integrity of the genome is checked. Vnutyuclear kinases ATR and ATM play an important role in the delay of the cell cycle in response to DNA damage. Intermediary proteins, such as Mre11 and MDC1, acquire post-translational modifications that are created by means of detector proteins. The intermediary proteins modified in this way then amplify the signal and transmit it to effector proteins such as Chk2, MDM2, and p53. The tumor suppressor protein p53 plays a crucial role in the cell cycle arrest or apoptosis after various stresses, including chemical damage to DNA. The main target of p53 is p21 (a cyclin-dependent kinase inhibitor), which causes the cell cycle to stop at the G1/G0 stage. The coordinated functioning of the protein groups described above and some others causes the cell cycle to stop, which makes possible to control DNA damage [29].

Classical methods of assessing genotoxicity reveal mutations associated with increased or loss of function, but are not informative enough in studying the molecular mechanisms of DNA damage. The MILLIPLEX^®^ MAP 7-plex DNA Damage/Genotoxicity Magnetic Bead kit allows to detect the expression and phosphorylation of a number of proteins involved in DNA damage, providing a faster and more accurate assessment of the state of the cell after exposure to genotoxic compounds. The 7-plex DNA Damage/Genotoxicity Magnetic Bead Kit is used to detect changes in phosphorylated Chk1 (Ser345), Chk2 (Thr68), H2AX (Ser139) and p53 (Ser15), as well as total protein levels of ATR, MDM2, and p21 in cellular lysates using the Luminex^®^ system. Figure 15 shows the data of multiplex analysis performed using the 7-plex DNA Damage/Genotoxicity Magnetic Bead Kit. The obtained results characterize the effect of the leading compound 7 on the DNA of M-HeLa cells at concentrations of IC_50_/2 and IC_50_. It is shown that the values of the average fluorescence intensity of the markers Chk1, Chk2, H2AX, MDM2, p53, and p21 significantly increase with increasing concentration compared to the control untreated compound **7** samples. The fluorescence intensity of OTR markers also increases relative to the reference sample. A high concentration of p21 protein may indicate the arrest of the M-HeLa cell cycle in the G1/G0 phase. This assumption was confirmed by cell cycle analysis using the fluorescent dye propidium iodide. Thus, the cytotoxic effect of the tested compounds can be characterized by the induction of apoptosis along the internal pathway associated with mitochondrial dysfunction, cell cycle delay in the G1/G0 phase.

## 3. Materials and Methods

### 3.1. Synthesis and Characterisation

All reagents (Acros Organics (Belgium), Alfa Aesar (USA)) were used without further purification. The 1,2,3,4-tetrahydropyrimidine-2-thions [22,23,24,25,26], ethyl (*Z*)-2-(2-hydroxybenzylidene)-7-methyl-3-oxo-5-phenyl-2,3-dihydro-5*H*-thiazolo[3,2-*a*]pyrimidine-6-carboxylate 7 [17], ethyl (*Z*)-2-(4-hydroxybenzylidene)-7-methyl-3-oxo-5-phenyl-2,3-dihydro-5*H*-thiazolo[3,2-*a*]pyrimidine-6-carboxylate 13 [30], ethyl (*Z*)-2-(2-hydroxybenzylidene)-5-(4-methoxyphenyl)-7-methyl-3-oxo-2,3-dihydro-5*H*-thiazolo[3,2-*a*]pyrimidine-6-carboxylate 8 [17] and ethyl (*Z*)-2-(2-hydroxybenzylidene)-5-(2-methoxyphenyl)-7-methyl-3-oxo-2,3-dihydro-5*H*-thiazolo[3,2-*a*]pyrimidine-6-carboxylate 9 [17] were synthesized according reported methods.

NMR experiments were performed on Bruker Avance instruments with an operating frequency of 400, 500, and 600 MHz for shooting ^1^H and ^13^C NMR spectra. Chemical shifts were determined relative to the signals of residual protons of the CDCl_3_ or DMSO-d_6_ solvents.

IR spectra in KBr tablets were recorded on a Bruker Vector-22.

Electrospray ionization (ESI) mass spectra were obtained using a Bruker AmaZon X ion trap mass spectrometer. Melting points were determined on a BOETIUS melting table with an RNMK 05 imaging device.

#### 3.1.1. General Method for Compounds **10**–**15** Preparation

The hydrochloride of the appropriate thiazolo[3,2-*a*]pyrimidine (1 mol) was mixed with CHCl_3_ (50 mL) and water (50 mL) solution containing 1 mol NaOH and stirred for 30 min at room temperature. Then, aromatic aldehyde (2- or 4-hydroxybenzaldehyde) (1 mol) and a catalytic amount (several drops) of pyrrolidine were added. A resulting mixture was stirred for 6 h under refluxing conditions. After cooling, the formed precipitate was filtered off, washed with ethanol, and purified by recrystallization from methanol followed by drying in vacuum for 2 h at 100 °C temperature affording a pure product.

Ethyl (*Z*)-2-(2-hydroxybenzylidene)-7-methyl-5-(3-nitrophenyl)-3-oxo-2,3-dihydro-5*H*-thiazolo[3,2-*a*]pyrimidine-6-carboxylate 10. Yield 87%, orange powder, mp 187–189 °C. IR (KBr, cm^−1^): 3272 (OH); 1704 (C=O); 1595, 1535 (C=N); 1158, 748 (C-S). ^1^H NMR (400 MHz, DMSO-d_6_, 25 °C) δ_H_ ppm: 1.11 (t, *J* = 7.1 Hz, 3H, OCH_2_CH_3_), 2.42 (s, 3H, CH_3_), 4.00–4.08 (m, 2H, OCH_2_CH_3_), 6.18 (s, 1H, CH-Ar), 6.94–6.98 (m, 2H, CH (Ar)), 7.30–7.38 (m, 2H, CH (Ar)), 7.66–7.71 (m, 1H, CH (Ar)), 7.78–7.80 (m, 1H, CH (Ar)), 7.97 (s, 1H, C=CH), 8.13 (s, 1H, CH (Ar)), 8.17–8.18 (m, 1H, CH (Ar)), 10.62 (br.s, 1H, OH). ^13^C NMR (100 MHz, DMSO-d_6_, 25 °C) δ_C_ ppm: 14.3, 23.1, 55.0, 60.7, 107.9, 116.7, 118.5, 120.1, 120.3, 120.4, 123.0, 123.9, 129.4, 131.0, 133.3, 134.7, 148.2, 156.8, 157.7, 165.1, 165.7. MS (ESI), *m*/*z*: 466 [M + H]^+^ (see Appendix A).

(*Z*)-6-benzoyl-2-(2-hydroxybenzylidene)-5-(2-methoxyphenyl)-7-methyl-5*H*-thiazolo[3,2-*a*]pyrimidin-3(2*H*)-one 11. Yield 45%, orange powder, mp 167–169 °C. IR (KBr, cm^−1^): 3448 (OH); 1624 (C=O); 1572 (C=N); 1487; 1272; 751 (C-S). ^1^H NMR (400 MHz, DMSO-d_6_, 25 °C) δ_H_ ppm: 1.70 (s, 3H, CH_3_), 3.62 (s, 3H, OCH_3_), 6.27 (s, 1H, CH-Ar), 6.85–6.88 (m, 1H, CH (Ar)), 6.95–7.00 (m, 4H, CH (Ar)), 7.21–7.24 (m, 1H, CH (Ar)), 7.32–7.35 (m, 1H, CH (Ar)), 7.40–7.48 (m, 3H, CH (Ar)), 7.58–7.61 (m, 3H, CH (Ar)), 7.94 (s, 1H, C=CH), 10.53 (s, 1H, OH). ^13^C NMR (100 MHz, DMSO-d_6_, 25 °C) δ_C_ ppm: 22.2, 54.2, 55.8, 112.2, 116.2, 116.6, 118.9, 120.3, 120.5, 121.0, 127.0, 127.6, 128.6, 128.7, 129.0, 129.3, 130.3, 132.9, 133.7, 138.4, 142.9, 154.7, 157.0, 157.6, 165.0, 195.8. MS (ESI), *m*/*z*: 483 [M + H]^+^ (see Appendix A).

Ethyl (*Z*)-5-(4-bromophenyl)-2-(2-hydroxybenzylidene)-7-methyl-3-oxo-2,3-dihydro-5*H*-thiazolo[3,2-*a*]pyrimidine-6-carboxylate 12. Yield 88%, orange powder, mp 201–203 °C. IR (KBr, cm^−1^): 3222 (OH); 1704 (C=O); 1595; 1553 (C=N); 1163. ^1^H NMR (500 MHz, DMSO-d_6_, 25 °C) δ_H_ ppm: 1.14 (t, *J* = 7.1 Hz, 3H, OCH_2_CH_3_), 2.39 (s, 3H, CH_3_), 4.02–4.09 (m, 2H, OCH_2_CH_3_), 6.03 (s, 1H, CH-Ar), 6.95–6.98 (m, 2H, CH (Ar)), 7.27 (d, *J* = 8.5 Hz, 2H, CH (Ar)), 7.31–7.38 (m, 2H, CH (Ar)), 7.56 (d, *J* = 8.5 Hz, 2H, CH (Ar)), 7.97 (s, 1H, C=CH). ^13^C NMR (100 MHz, DMSO-d_6_, 25 °C) δ_C_ ppm: 14.9, 23.5, 55.4, 61.2, 109.0, 117.2, 119.1, 120.7, 120.7, 120.8, 122.8, 129.7, 130.8, 132.6, 133.7, 140.7, 152.7, 157.0, 158.3, 165.5, 165.7. MS (ESI), *m*/*z*: 499, 501 [M + H]^+^ (see Appendix A).

Ethyl (*Z*)-5-(4-bromophenyl)-2-(4-hydroxybenzylidene)-7-methyl-3-oxo-2,3-dihydro-5*H*-thiazolo[3,2-*a*]pyrimidine-6-carboxylate 14. Yield 88%, orange powder, mp 207–209 °C. IR (KBr, cm^−1^): 3413 (OH); 1717 (C=O); 1585 (C=N); 1515; 1160. ^1^H NMR (600 MHz, DMSO-d_6_, 25 °C) δ_H_ ppm: 1.13 (t, *J* = 7 Hz, 3H, OCH_2_CH_3_), 2.38 (s, 3H, CH_3_), 4.03–4.08 (m, 2H, OCH_2_CH_3_), 6.01 (s, 1H, CH-Ar), 6.92 (d, *J* = 8.4 Hz, 2H, CH (Ar)), 7.25 (d, *J* = 8 Hz, 2H, CH (Ar)), 7.46 (d, *J* = 8.4 Hz, 2H, CH (Ar)), 7.54 (d, *J* = 8 Hz, 2H, CH (Ar)), 7.69 (s, 1H, C=CH). ^13^C NMR (100 MHz, DMSO-d_6_, 25 °C) δ_C_ ppm: 14.4, 22.8, 54.8, 60.7, 108.3, 115.4, 116.9, 122.2, 124.1, 130.2, 132.1, 133.1, 134.3, 140.3, 152.3, 156.6, 160.8, 165.0, 165.3. MS (ESI), *m*/*z*: 499, 501 [M + H]^+^ (see Appendix A).

Ethyl (*Z*)-2-(4-hydroxybenzylidene)-7-methyl-5-(3-nitrophenyl)-3-oxo-2,3-dihydro-5*H*-thiazolo[3,2-*a*]pyrimidine-6-carboxylate 15. Yield 84%, orange powder, mp 190–192 °C. IR (KBr, cm^−1^): 3048 (OH); 1717 (C=O); 1585; 1515; 1160. ^1^H NMR (500 MHz, DMSO-d_6_, 25 °C) δ_H_ ppm: 1.11 (t, *J* = 7.1 Hz, 3H, OCH_2_CH_3_), 2.42 (s, 3H, CH_3_), 3.99–4.08 (m, 2H, OCH_2_CH_3_), 6.17 (s, 1H, CH-Ar), 6.92 (d, *J* = 8.7 Hz, 2H, CH (Ar)), 7.47 (d, *J* = 8.7 Hz, 2H, CH (Ar)), 7.66–7.69 (m, 1H, CH (Ar)), 7.70 (s, 1H, C=CH), 7.77–7.79 (m, 1H, CH (Ar)), 8.12–8.13 (m, 1H, CH (Ar)), 8.16–8.18 (m, 1H, CH (Ar)), 10.39 (br.s, 1H, OH). ^13^C NMR (100 MHz, DMSO-d_6_, 25 °C) δ_C_ ppm: 14.3, 23.1, 55.0, 60.7, 107.7, 115.3, 116.9, 122.9, 123.9, 124.1, 131.0, 133.1, 134.5, 134.6, 142.8, 148.1, 153.1, 156.9, 160.9, 165.1. MS (ESI), *m*/*z*: 466 [M + H]^+^ (see Appendix A).

#### 3.1.2. Crystallization Conditions

Crystals of **11**, **12**, **13,** and **14** suitable for X-ray diffraction study were obtained by slow evaporation of ethanol solution (25 mL) containing 0.02 mol of dissolved compound after 5 days.

Crystals of solvates **12-DMSO** and **14-DMSO** suitable for X-ray diffraction study were obtained by slow evaporation of DMSO solution (10 mL) containing 0.02 mol of dissolved compound after 7 days.

Crystals of solvate **10-MeOH** suitable for X-ray diffraction study were obtained by slow evaporation of MeOH solution (15 mL) containing 0.02 mol of the dissolved compound after 3 days.

#### 3.1.3. Single Crystal X-ray Diffraction

The X-ray diffraction study of 12 was carried out at the “Belok/XSA” beamline of the Kurchatov Synchrotron Radiation Source [31,32]. Diffraction patterns were collected using Mardtb goniometer (marXperts GmbH, Werkstraße 3, 22844 Norderstedt, Germany) equipped with Rayonix SX165 CCD (Rayonix LLC., 1880 Oak Ave UNIT 120, Evanston, IL 60201, USA) 2D positional sensitive CCD detector (λ = 0.7450 Å, φ-scanning in 1.0° steps). All data were collected at 100(2) K.

X-ray diffraction analysis of 10-MeOH, 11, 12-DMSO, 13, 14, and 14-DMSO was performed on a Bruker D8 QUEST automatic three-circle diffractometer with a PHOTON III two-dimensional detector and an IμS DIAMOND microfocus X-ray tube (λ[Mo Kα] = 0.71073 Å) at cooling conditions. Data collection and processing of diffraction data were performed using APEX3 software package.

All structures were solved by the direct method using the SHELXT program [33] and refined by the full-matrix least squares method over F^2^ using the SHELXL program [34]. All calculations were performed in the WinGX software package [35], the calculation of the geometry of molecules and intermolecular interactions in crystals was carried out using the PLATON program [36], the drawings of molecules were performed using the ORTEP-3 [35] and MERCURY [37] programs.

Non-hydrogen atoms were refined in the anisotropic approximation. The positions of the hydrogen atoms H(O) were determined using difference Fourier maps, and these atoms were refined isotropically. The remaining hydrogen atoms are placed in geometrically calculated positions and included in the refinement in the “riding” model. The crystal of compound **9**, **12,** and **14** is a solvate with DMSO (1:1); crystal of compound **9**—solvate with methanol (1:1). Crystallographic data of structures **9**–**14** were deposited at the Cambridge Crystallographic Data Center, registration numbers and the most important characteristics are given in Table 3.

### 3.2. Biological Study

#### 3.2.1. Cells and Materials

For the experiments, tumor cell cultures of M-HeLa clone 11 (epithelioid carcinoma of the cervix, subline HeLa., clone M-HeLa), HuTu 80, human duodenal adenocarcinoma, MCF7—human breast adenocarcinoma (pleural fluid) collected from Institute of Cytology, Russian Academy of Sciences (St. Petersburg, Russia); PC3—prostate adenocarcinoma cell line collected from ATCC (American Type Cell Collection, Manassas, VA, USA; CRL 1435; human liver cells (Chang liver) and the Research Institute of Virology of the Russian Academy of Medical Sciences (Moscow, Russia) were used for cytotoxicity analysis.

#### 3.2.2. MTT Assay

The cytotoxic effect on cells was determined using the colorimetric method of cell proliferation—the MTT test. NADP-H-dependent cellular oxidoreductase enzymes can, under certain conditions, reflect the number of viable cells. These enzymes are able to reduce the tetrazolium dye (MTT)—3-(4,5-dimethylthiazol-2-yl)-2,5-diphenyl-tetrazolium bromide to insoluble blue-violet formazan, which crystallizes inside the cell. The amount of formazan formed is proportional to the number of cells with active metabolism. Cells were seeded on a 96-well Nunc plate at a concentration of 5 × 10^3^ cells per well in a volume of 100 μL of medium and cultured in a CO_2_ incubator at 37 °C until a monolayer was formed. Then the nutrient medium was removed and 100 µL of solution of the test drug in the given dilutions was added to the wells, which were prepared directly in the nutrient medium with the addition of 5% DMSO to improve solubility. After 48 h of incubation of the cells with the tested compounds, the nutrient medium was removed from the plates and 100 µL of the nutrient medium without serum with MTT at a concentration of 0.5 mg/mL was added and incubated for 4 h at 37 °C. Formazan crystals were added to 100 µL of DMSO. Optical density was recorded at 540 nm on an Invitrologic microplate reader (Novosibirsk, Russia). The experiments for all compounds were repeated three times.

#### 3.2.3. Induction of Apoptotic Effects by Test Compounds

##### Flow Cytometry Assay

Cell Culture. M-HeLa cells at 1 × 10^6^ cells/well in a final volume of 2 mL were seeded into six-well plates. After 48 h of incubation, various concentrations of compound **7** were added to the wells.

Cell Apoptosis Analysis. The cells were harvested at 2000 rpm for 5 min and then washed twice with ice-cold PBS, followed by resuspension in binding buffer. Next, the samples were incubated with 5 μL of annexin V-Alexa Fluor 647 (Sigma-Aldrich, St. Louis, MO, USA) and 5 μL of propidium iodide for 15 min at room temperature in the dark. Finally, the cells were analyzed by flow cytometry (Guava easy Cyte, MERCK, Kenilworth, NJ, USA) within 1 h. The experiments were repeated three times.

Mitochondrial Membrane Potential. Cells were harvested at 2000 rpm for 5 min and then washed twice with ice-cold PBS, followed by resuspension in JC-10 (10 µg/mL) and incubation at 37 °C for 10 min. After the cells were rinsed three times and suspended in PBS, the JC-10 fluorescence was observed by flow cytometry (Guava easy Cyte, MERCK, Kenilworth, NJ, USA).

#### 3.2.4. Detection of Intracellular ROS

M-HeLa cells were incubated with compound **7** at concentrations of IC_50_/2 and IC_50_ for 48 h. ROS generation was investigated using flow cytometry assay and CellROX^®^ Deep Red flow cytometry kit. For this, M-HeLa cells were harvested at 2000 rpm for 5 min and then washed twice with ice-cold PBS, followed by resuspension in 0.1 mL of medium without FBS, to which 0.2 μL of CellROX^®^ Deep Red was added and incubated at 37 °C for 30 min. After three times washing the cells and suspending them in PBS, the production of ROS in the cells was immediately monitored using flow cytometer (Guava easy Cyte, MERCK, Kenilworth, NJ, USA).

#### 3.2.5. Multiplex Analysis of Markers DNA Damage/Genotoxicity

The studies were carried out according to the standard protocol. M-HeLa cells were incubated for 24 h with the test substance. Cells were lysed in MILLIPLEX^®^ MAP Lysis buffer containing protease inhibitors. A total of 20 μg of total protein of each lysate diluted in MILLIPLEX^®^ MAP Assay Buffer 2 was analyzed according to the analysis protocol (the lysate was incubated at 4 °C overnight). The mean fluorescence intensity (MFI) was detected using Luminex^®^ system, MERCK, Kenilworth, NJ, USA.

##### Statistical Analysis

The IC_50_ values were calculated using the online calculator MLA-Quest Graph™ IC_50_ Calculator AAT Bioquest, Inc., 14 February 2021. Statistical analysis was performed using the Mann–Whitney test (*p* < 0.05). Tabular and graphical data contain the averages and standard error.

## 4. Conclusions

Synthesis of new 2-(2- and 4-hydroxybenzylidene)thiazolo[3,2-*a*]pyrimidine derivatives **10**–**15** containing phenyl, *m*-nitrophenyl, *p*-bromophenyl, and *o*-anisyl substituents at the C5 atom was successfully achieved in high yields. The single crystal X-ray diffraction study revealed that the supramolecular motif in the crystalline phase of the obtained compounds can be controlled by adjusting the interplay between various types of non-covalent interactions such as H-, Br–π, O–π, π–π bonding through the rational choice of the substituents at C2, C5, and C6 atoms. Depending on the used 2-or 4-hydroxybenzylidene derivatives, the self-assembly pathway leads to the formation of racemic H-bonded dimers (**11**, **12**) or infinite 1D homochiral chains (**13**, **14**), respectively, which can be explained by the close disposition of H-bond donor and H-bond acceptor observed for **11** and **12**, and relatively high H-accepting ability of N8 pyrimidine atom in the case of **13** and **14**. Moreover, the 1D chains can be generated for 2-hydroxybenzylidene by involving protic EtOH or MeOH acting as bridges in intermolecular H-bonding between the obtained heterocycles (**10-MeOH**). It was demonstrated that upon crystal packing, the relatively weak Br–π, O–π, and π–π interactions also play a significant role which can be enhanced by switching off the intermolecular H-bonding when DMSO molecules are involved in coordination with OH-donor group.

A series composed of new compounds 2-(2- and 4-hydroxybenzylidene)thiazolo[3,2-*a*]pyrimidine derivatives **10**–**15** and their earlier reported analogues **7**–**9** were tested in vitro as antitumor agents. The studied compounds showed high or moderate activity against a number of cancer lines of various genesis and demonstrated moderate cytotoxicity against normal liver cells. The most significant results of the studied compounds were shown in relation to the cell lines of cervical carcinoma (M-HeLa) and human duodenal adenocarcinoma (HuTu 80). Their cytotoxic effect was confirmed by comparing with drug sorafenib, and compound **7** has demonstrated two times higher efficiency. According to the calculated indices of selectivity of cytotoxic action, compound **7** also demonstrates better selectivity with respect to the M-HeLa and HuTu 80cell lines. Using the flow cytometry method and multiplex analysis of DNA damage/genotoxicity markers, it was shown that the cytotoxic effect of the tested compound **7** can be characterized by the induction of apoptosis along the internal pathway associated with mitochondrial dysfunction, cell cycle delay in the G1/G0 phase. The design of new thiazolo[3,2-*a*]pyrimidines exhibiting higher anti-tumor activity with simultaneously low normal cells toxicity is in progress.

## Figures and Tables

**Figure 1 molecules-27-07747-f001:**
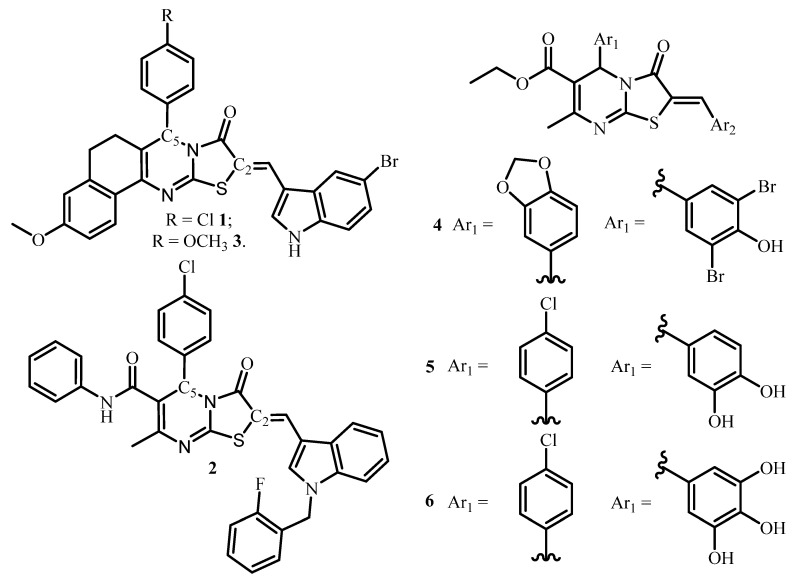
Thiazolo[3,2-*a*]pyrimidine compounds used as promising enzyme inhibitors and antitumor drugs.

**Figure 2 molecules-27-07747-f002:**
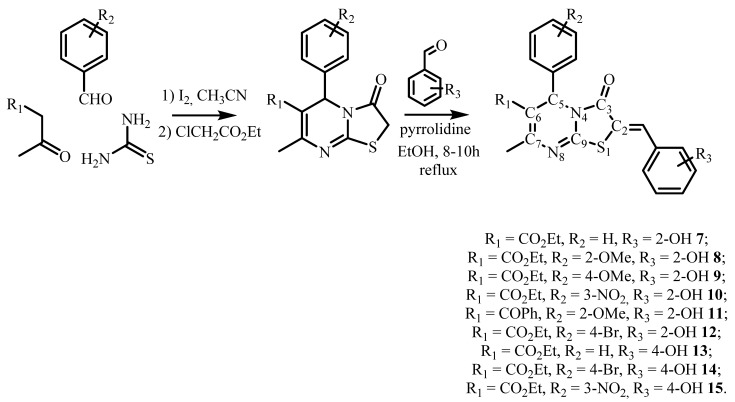
Synthesis of 2-and 4-(hydroxybenzylidene)thiazolo[3,2-*a*]pyrimidines **7**–**15**.

**Figure 3 molecules-27-07747-f003:**
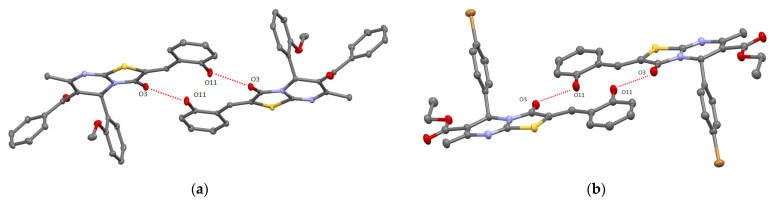
ORTEP-view of supramolecular dimers resulting from intermolecular H-bonding in crystals of compound **11** (**a**) and **12** (**b**). The ellipsoids are presented with 50% probability, H-atoms are omitted for clarity. H-bonding is presented by red dotted lines.

**Figure 4 molecules-27-07747-f004:**
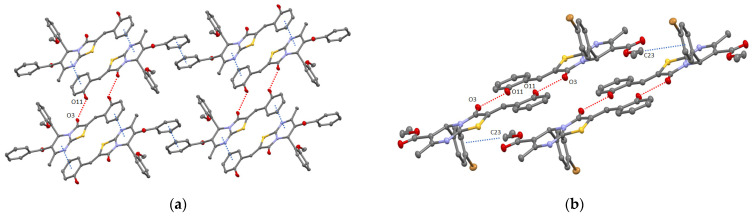
ORTEP view of crystal packing of **11** (**a**) and **12** (**b**) showing 2D layer and 1D chain formation resulting from both the intermolecular H-bonding and/or π–π/CH–π interactions (red and blue dotted lines, respectively). The ellipsoids are presented with 50% probability, H-atoms are omitted for clarity.

**Figure 5 molecules-27-07747-f005:**
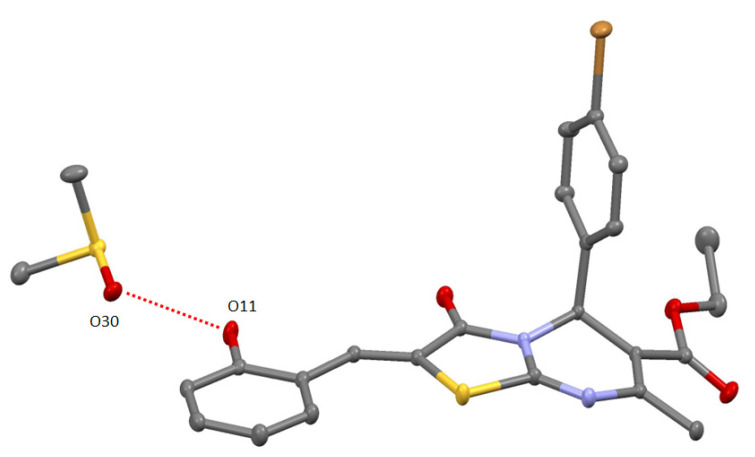
ORTEP-view of solvate complex **12-DMSO**. The ellipsoids are presented with 50% probability, H-atoms are omitted for clarity. H-bonding is presented by red dotted lines.

**Figure 6 molecules-27-07747-f006:**
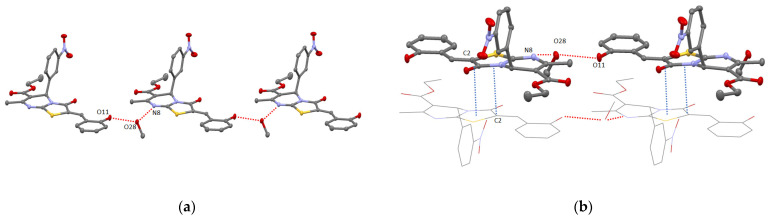
ORTEP view of supramolecular 1D chains resulting from intermolecular H-bonding in the crystals of compound **10-MeOH** (**a**) and π-staking involving the thiazolyl groups (**b**). The ellipsoids are presented with 50% probability, H-atoms are omitted for clarity. H-bonding and π-stacking are presented by red and blue dotted lines, respectively.

**Figure 7 molecules-27-07747-f007:**
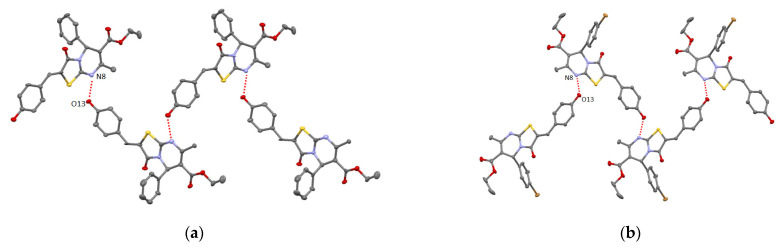
ORTEP view of supramolecular 1D homochiral chains formation of **13** (**a**) and **14** (**b**). The ellipsoids are presented with 50% probability, H-atoms are omitted for clarity.

**Figure 8 molecules-27-07747-f008:**
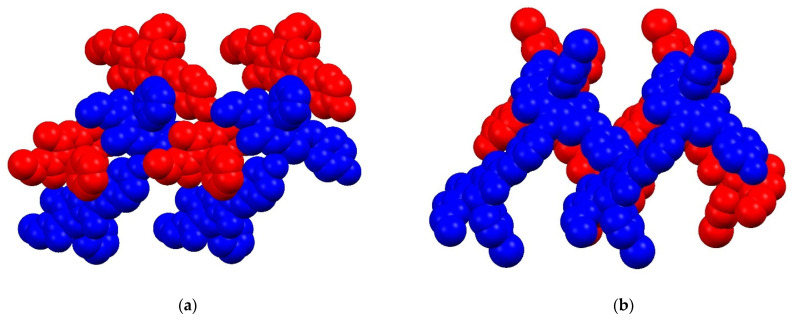
A portion of crystal packing of R- and S-homochiral chains (colored in blue and red, respectively) for **13** (**a**) and **14** (**b**).

**Figure 9 molecules-27-07747-f009:**
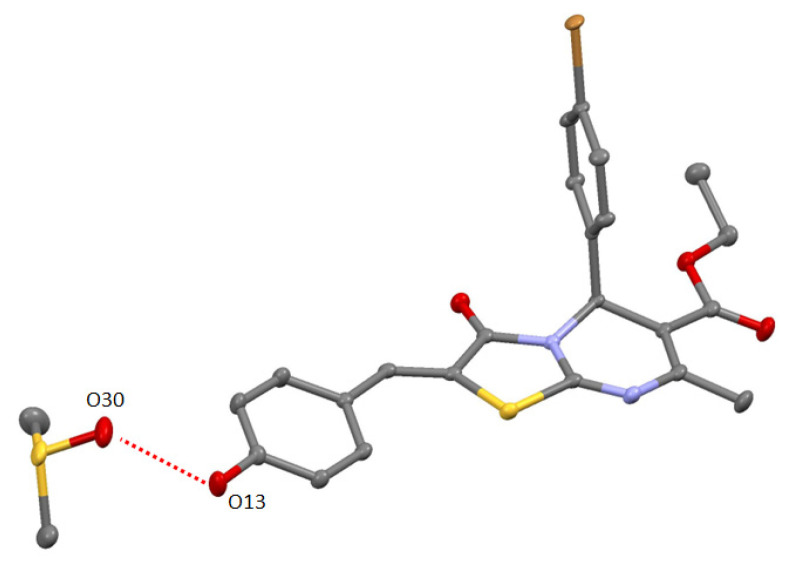
Crystal structure of solvate complex **14-DMSO**.

**Figure 10 molecules-27-07747-f010:**
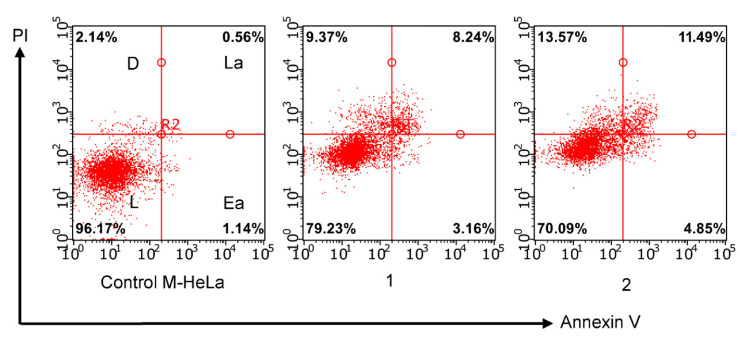
Induction of apoptosis in M-HeLa cells incubated with compound **7**. 1—at concentration IC_50_/2 (6 µM); 2—at concentration IC_50_ (12 µM); L—living cells; D—dead cells; Ea—early apoptotic cells; La—late apoptotic cells.

**Figure 11 molecules-27-07747-f011:**
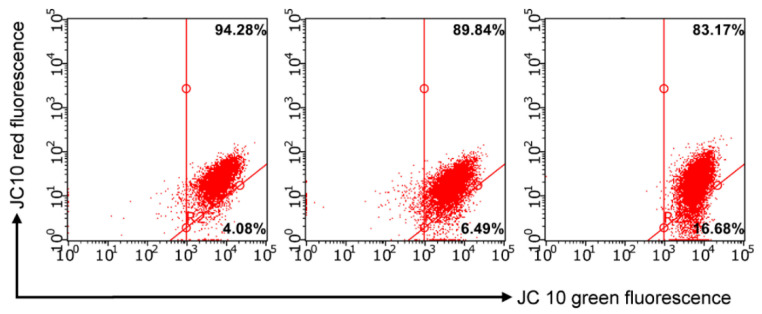
Effects on the mitochondrial membrane potential by **7** in M-HeLa cells. 1—at concentration IC_50_/2 (6 µM); 2—at concentration IC_50_ (12 µM).

**Figure 12 molecules-27-07747-f012:**
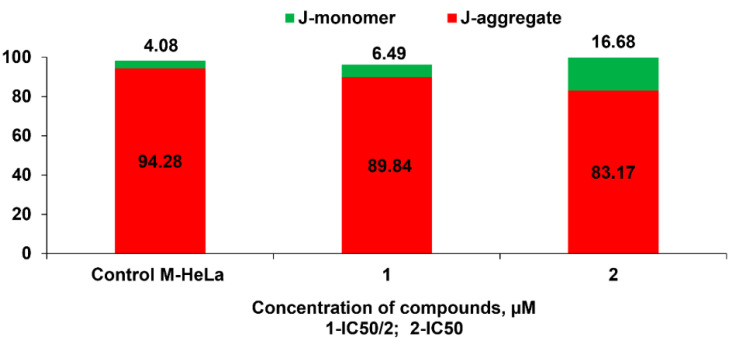
Quantitative determination of % M-HeLa cells with red and green aggregates. 1—at concentration IC_50_/2 (6 µM); 2—at concentration IC_50_ (12 µM).

**Figure 13 molecules-27-07747-f013:**
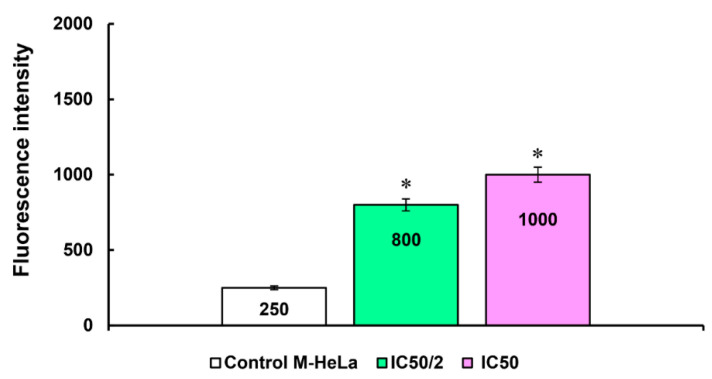
Induction of ROS production by compound **7**. 1—at concentration IC_50_/2 (6 µM); 2—at concentration IC_50_ (12 µM). * Values indicate *p* < 0.05.

**Figure 14 molecules-27-07747-f014:**
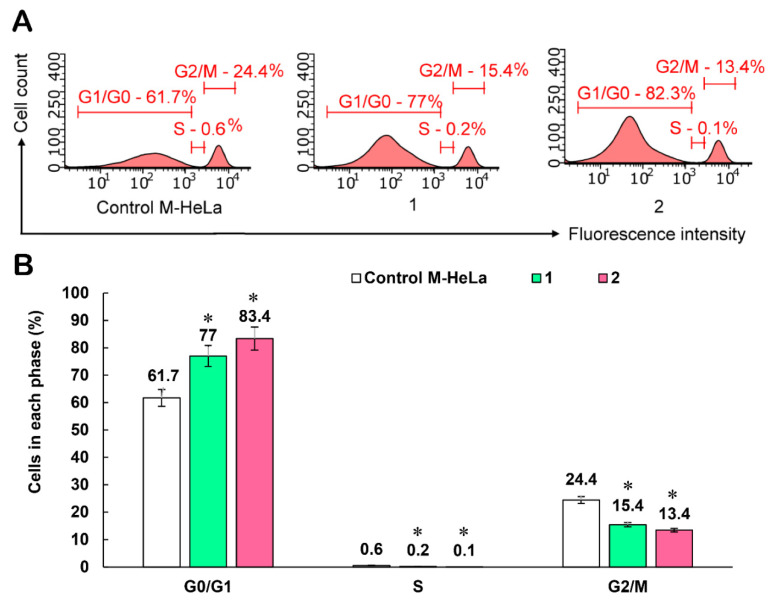
Effect of compound **7** on M-HeLa cell cycle arrest. 1—at concentration IC_50_/2 (6 µM); 2—at concentration IC_50_ (12 µM); (**A**) cell distribution histograms. (**B**) Percentage of cells in the G0/G1, S, and G2/M phases (data are presented as mean ± SD of three independent experiments). * Values indicate *p* < 0.05.

**Figure 15 molecules-27-07747-f015:**
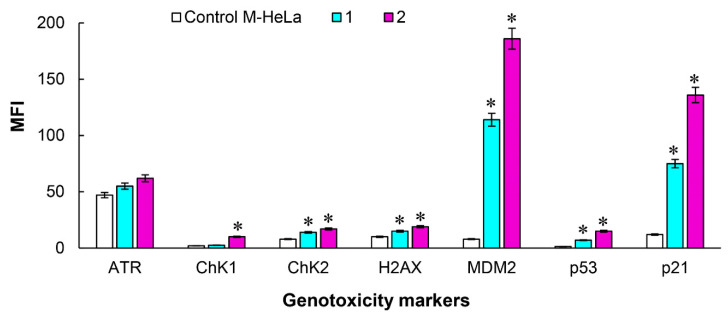
Multiplex analysis of markers DNA damage/genotoxicity in M-HeLa cells treated compound **7**. 1—at concentration IC_50_/2 (6 µM); 2—at concentration IC_50_ (12 µM). M-HeLa cells untreated with the test substance (control). Data are presented as mean ± SD of three independent experiments * Values indicate *p* < 0.05.

**Table 1 molecules-27-07747-t001:** Cytotoxic effects (IC_50_, µM) of test-compounds.

Test Compounds	IC_50_ (µM)				
	Cancer Cell Lines				Normal Cell Lines
	M-HeLa ^a^	MCF-7 ^b^	PC3 ^c^	HuTu 80 ^d^	Chang Liver (HeLa) ^e^
7	11.9 ± 0.9	43.7 ± 3.5	77.6 ± 6.7	10.2 ± 0.8	75.0 ± 5.7
8	90 ± 8.3	93.3 ± 8.2	93.7 ± 8.5	85.9 ± 8.0	56.4 ± 4.5
9	20.6 ± 1.7	32.1 ± 2.5	23.3 ± 1.9	26.8 ± 2.1	22.7 ± 1.8
10	55.0 ± 4.4	56.0 ± 4.5	56.0 ± 4.4	60 ± 4.7	88.0 ± 7.0
11	56.4 ± 4.5	81.0 ± 7.4	54.0 ± 4.3	39.0 ± 3.1	64.0 ± 5.1
12	18.8 ± 1.5	22.3 ± 1.4	28.1 ± 2.2	23.7 ± 1.8	78.2 ± 6.2
13	70.0 ± 5.5	>100	54.0 ± 4.3	43.3 ± 3.5	86.1 ± 6.8
14	31.0 ± 2.4	53.3 ± 4.2	49.3 ± 3.9	32.1 ± 2.5	53.2 ± 4.2
15	>100	>100	70.0 ± 5.5	74.5 ± 5.7	91.4 ± 7.2
Sorafenib 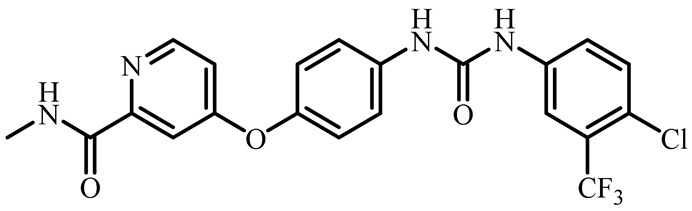	25.0 ± 1.8	27.5 ± 2.2	12.7 ± 1.1	13.0 ± 1.2	21.7 ± 1.7

The experiments were performed in triplicate. Results are expressed as the mean ± standard deviation (SD); ^a^ M-HeLa clone 11—epithelioid carcinoma of the cervix, subline HeLa, clone M-HeLa; ^b^ MCF-7 is an epithelial-like cell line derived from invasive human mammary duct adenocarcinoma; ^c^ PC3—prostate adenocarcinoma cell line from ATCC (Collection of American-type cells, USA); ^d^ HuTu 80—human duodenal adenocarcinoma from the collection of the Institute of Cytology of the Russian Academy of Sciences (St. Petersburg); ^e^ Chang liver (HeLa derivative)—human liver cells from the collection of the Research Institute of Virology of the Russian Academy of Medical Sciences (Moscow).

**Table 2 molecules-27-07747-t002:** Cytotoxic effects (μM) and selectivity index values (SI) of lead compounds.

Test Compounds	Normal Cell Lines								
	M-HeLa ^a^		MCF-7 ^b^		PC3 ^c^		HuTu 80 ^d^		Chang Liver (HeLa) ^e^
	IC_50_	SI	IC_50_	SI	IC_50_	SI	IC_50_	SI	
7	11.9 ± 0.9	6.3	43.7 ± 3.5	1.7	77.6 ± 6.7	ns	10.0 ± 0.8	7.5	75.0 ± 5.7
9	20.6 ± 1.7	1.1	32.1 ± 2.5	ns	23.3 ± 1.9	ns	26.8 ± 2.1	ns	22.7 ± 1.8
10	55.0 ± 4.4	1.6	56.0 ± 4.5	1.6	56.0 ± 4.4	1.6	14.6 ± 0.3	6.0	88.0 ± 7.0
12	18.8 ± 1.5	4.2	52.7 ± 4.2	1.5	28.1 ± 2.2	2.8	23.7 ± 1.8	3.3	78.2 ± 6.2
Sorafenib	25.0 ± 1.8	ns	27.5 ± 2.2	ns	27.5 ± 2.2	ns	13.0 ± 1.2	1.7	21.7 ± 1.7

The experiments were performed in triplicate. Results are expressed as the mean ± standard deviation (SD); ns—no selectivity; ^a^ M-HeLa clone 11—epithelioid carcinoma of the cervix, subline HeLa, clone M-HeLa; ^b^ MCF-7 is an epithelial-like cell line derived from invasive human mammary duct adenocarcinoma; ^c^ PC3—prostate adenocarcinoma cell line from ATCC (Collection of American-type cells, USA); ^d^ HuTu 80—human duodenal adenocarcinoma from the collection of the Institute of Cytology of the Russian Academy of Sciences (St. Petersburg); ^e^ Chang liver (HeLa derivative)—human liver cells from the collection of the Research Institute of Virology of the Russian Academy of Medical Sciences (Moscow).

**Table 3 molecules-27-07747-t003:** Crystallographic data for studied compounds.

Compound	10-MeOH	11	12	12-DMSO
Molecular formula	C_23_H_19_N_2_O_5_S, CH_4_O	C_28_H_22_N_2_O_4_S	C_23_H_19_BrN_2_O_4_S	C_23_H_19_BrN_2_O_4_S, C_2_H_6_OS
Sum Formula	C_24_H_23_N_2_O_5_S	C_28_H_22_N_2_O_4_S	C_23_H_19_BrN_2_O_4_S	C_26_H_28_N_2_O_6_S_2_
Formula Weight	497.51	482.54	499.36	577.49
Crystal System	triclinic	triclinic	triclinic	triclinic
Space group	*P-1 (P1bar)*	*P-1 (P1bar)*	*P-1 (P1bar)*	*P-1 (P1bar)*
Temp. of measurement, K	100(2)	105(2)	100(2)	105(2)
Cell parameters	*a* = 9.4229(5) Å, *b* = 9.4636(5) Å, *c* = 12.8810(7) Å; α = 82.651(2)° β = 89.551(2)° γ = 77.443(2)°	*a* = 9.7165(8) Å, *b* = 10.1644(9) Å, *c* = 13.4602(12) Å; α = 101.395(3)° β = 99.097(3)° γ = 113.423(3)°	*a* = 7.7700(16) Å, *b* = 11.950(2) Å, *c* = 12.540(3) Å; α = 64.59(3)° β = 82.44(3)° γ = 86.70(3)°	*a* = 8.5364(2) Å, *b* = 11.8152(3) Å, *c* = 13.4137(4) Å; α = 103.604(1)° β = 106.947(1)° γ = 92.508(1)°
V [Å^3^]	1111.74(10)	1153.65(18)	1042.6(5)	1248.63(6)
*Z* and *Z*′	2 and 1	2 and 1	2 and 1	2 and 1
D(calc) [g/cm^3^]	1.486	1.389	1.591	1.536
λ (Å)	(MoKα) 0.71073	(MoKα) 0.71073	0.7450 (synchrotron)	(MoKα) 0.71073
μ [/mm]	0.199	0.180	2.355	1.853
*F*(000)	520	504	508	592
Theta Min-Max [Deg]	2.2–28.0°	2.3–30.0°	1.9–31.0°	1.8–32.0°
Reflections measured	41,825	89,193	20,991	44,055
Independent reflections	5357	6713	5717	8644
Observed reflections [*I* > 2σ(*I*)]	4506	5883	5297	7568
Goodness of fit	1.138	1.030	1.076	1.048
*R* [*I* > 2σ(*I*)]	*R*1 = 0.0391, *wR*2 = 0.1137	*R*1 = 0.0340, *wR*2 = 0.0956	*R*1 = 0.0326, *wR*2 = 0.0867	*R*1 = 0.0356, *wR*2 = 0.1004
*R* (all reflections)	*R*1 = 0.0491, *wR*2 = 0.1189	*R*1 = 0.0397, *wR*2 = 0.0985	*R*1 = 0.0361, *wR*2 = 0.0901	*R*1 = 0.0418, *wR*2 = 0.1036
Max. and Min. Resd. Dens. [e/Å^−3^]	0.52 and −0.23	0.46 and −0.27	0.44 and −0.86	1.19 and −0.51
Depositor numbers in CCDC	2,213,096	2,213,093	2,213,097	2,213,094
Compound	**13**	**14**	**14-DMSO**	
Molecular formula	C_23_H_20_N_2_O_4_S	C_23_H_19_BrN_2_O_4_S	C_23_H_19_BrN_2_O_4_S, C_2_H_6_OS	
Sum Formula	C_23_H_20_N_2_O_4_S	C_23_H_19_BrN_2_O_4_S	C_26_H_28_N_2_O_6_S_2_	
Formula Weight	420.47	499.36	577.49	
Crystal System	monoclinic	monoclinic	triclinic	
Space group	*Pc*	*P2_1_*/*n*	*P-1 (P1bar)*	
Temp. of measurement, K	110(2)	110(2)	105(2)	
Cell parameters	*a* = 11.0059(15) Å, *b* = 13.6530(15) Å, *c* = 14.8440(18) Å; β = 109.743(3)°	*a* = 11.3394(5) Å, *b* = 14.2710(7) Å, *c* = 13.6906(6) Å; β = 104.046(2)°	*a* = 8.8459(4) Å, *b* = 11.2404(4) Å, *c* = 12.8542(5) Å; α = 89.785(2)° β = 88.746(2)° γ = 83.140(2)°	
V [Å^3^]	2099.4(4)	2149.24(17)	1268.66(9)	
*Z* and *Z*′	4 and 2	4 and 1	2 and 1	
D(calc) [g/cm^3^]	1.330	1.543	1.512	
λ (Å)	(MoKα) 0.71073	(MoKα) 0.71073	(MoKα) 0.71073	
μ [/mm]	0.186	2.043	1.824	
*F*(000)	880	1016	592	
Theta Min-Max [Deg]	1.5–26.6°	2.1–30.0°,	1.8–30.0°	
Reflections measured	69,218	104,197	89,171	
Independent reflections	8626	6257	7381	
Observed reflections [*I* > 2σ(*I*)]	6008	4741	6684	
Goodness of fit	1.062	1.009	1.027	
*R* [*I* > 2σ(*I*)]	*R*1 0.0576, *wR*2 = 0.1249	*R*1 = 0.0347, *wR*2 = 0.0713	*R*1 = 0.0304, *wR*2 = 0.0761	
*R* (all reflections)	*R*1 = 0.1017, *wR*2 = 0.1417	*R*1 = 0.0578, *wR*2 = 0.0783	*R*1 = 0.0355, *wR*2 = 0.0786	
Max. and Min. Resd. Dens. [e/Å^−3^]	0.35 and −0.55	0.55 and −0.41	0.68 and −0.61	
Flack parameter	−0.35(8) (twin, BASF 0.52081)	-	-	
Depositor numbers in CCDC	2,213,099	2,213,095	2,213,098	

## Data Availability

The data presented in this study are contained within the article or in Appendix A, or are available on request from the corresponding author Igor Antipin.

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
