# Peer review of "Synthesis, Self-Assembly in Crystalline Phase and Anti-Tumor Activity of 2-(2-/4-Hydroxybenzylidene)thiazolo[3,2-a]pyrimidines"

_molecules, 2022, doi:10.3390/molecules27227747_

Round 1
Reviewer 1 Report
In the manuscript of Artem S. Agarkov and co-workers, a series of new thiazolo[3,2-a]pyrimidine derivatives with different substituents have been synthesized and fully characterized. The obtained thiazolo[3,2-a]pyrimidines showed high or moderate activity against different tumor cells.
As the continuous research of Artem S. Agarkov group, this work revealed the role of intermolecular H-bonding in formation of supramolecular architectures of these thiazolo[3,2-a]pyrimidines derivatives via the SCXRD study. And the antitumor activities were studied.
Regarding the methodology, all the R1 substituents are electron-withdrawing groups (CO2Et/COPh), and all the R3 substituents are electron-donating groups (2/4-OH). Maybe more different substituents should be carefully considered.
The conclusions are consistent with the evidence and arguments presented and address the main question posed. The references are appropriate.
Although the paper is very well written and packed a large amount of information, some typographical and drawing errors were identified, and a careful check by an expert proofreader is recommended. Such as,
Figure 2 should be clearly readable, such as Figure 2.
Line 164, “methanole” should be “methanol”.
I recommend publication after minor modifications.
Author Response
We are grateful to the reviewers for their constructive remarks, comments and suggestions. We've revised our manuscript according to their recommendations. The changes are highlighted in yellow in the revised version of the manuscript. Below is also our detailed point-by-point response.
Comments and Suggestions from Reviewer 1.
Regarding the methodology, all the R1 substituents are electron-withdrawing groups (CO2Et/COPh), and all the R3 substituents are electron-donating groups (2/4-OH). Maybe more different substituents should be carefully considered.
We do appreciate the referee remark. We are currently working on the preparation of thiazolopyrimidine derivatives containing other types of electron donor and withdrawing groups. We plan to present these results in our next papers.
Figure 2 should be clearly readable, such as Figure 2.
We thank referee for his/her comment. The appropriate correction has been done in the revised version of the manuscript.
Line 164, “methanole” should be “methanol”.
We thank referee for his/her comment. The appropriate correction has been done in the revised version of the manuscript.
Reviewer 2 Report
In the article by Agarkov et al. reported synthesis and antitumor activity of 2-substituted thiazolo[3,2-a]pyrimidine derivatives. All compounds are fairly well characterized by a number of methods: 1H and 13C NMR, IR, mass-spectra and single crystal X-ray diffraction. Considering the novelty and utility of this study, this manuscript should be accepted in Molecules after addressing following comments.
1. It would be useful to assign at least the strongest bands in the IR spectra
2. In the “4.1. Synthesis and characterization” there is information about two-dimensional and low temperature NMR spectra that are missing in the manuscript, please correct.
3. Structure of Sorafenib is missing in the manuscript. It may be added for readers convenience.
4. There are missing footnotes in the Table 2 (a, b, c, d, e)
5. Is Chang liver correct name or it should be clarified as Chang liver (HeLa)?
6. It appears that not all of the synthesized substances are dry. Please note that the peak height ratio (I can't integrate from the picture) of water and DMSO varies from compound to compound. It looks like compound 12 contain much more water than compound 14.
7. Analytical data and spectra (in the SI) of compound 13 are missing.
In supporting info:
8. Russian letters on NMR spectra axis labels
9. It seems that solubility of compound 10 in DMSO-d6 it rather low, 1H NMR spectrum of compound 10 is quite noisy, 13C NMR spectrum is even worse and S/N ratio is barely acceptable. Is it possible to find a more suitable deuterated solvent? Or acquire a spectrum at a higher temperature in DMSO, maybe the solubility will increase enough.
10. The same about compound 15
11. Also 1H NMR spectrum of compound 10 contains unidentified peak at 2.09 ppm. All NMR peaks should be labeled (water, dmso-d5/d6)
Author Response
We are grateful to the reviewers for their constructive remarks, comments and suggestions. We've revised our manuscript according to their recommendations. The changes are highlighted in yellow in the revised version of the manuscript. Below is also our detailed point-by-point response.
Comments and Suggestions for Reviewer 2
- It would be useful to assign at least the strongest bands in the IR spectra
We thank referee for his/her comment. We assigned the most characteristic signals in the IR spectra in the revised version of the Supplementary Material file.
- In the “4.1. Synthesis and characterization” there is information about two-dimensional and low temperature NMR spectra that are missing in the manuscript, please correct.
We thank referee for his/her remark. The notice about 2D and low temperature NMR spectroscopy studies were removed from the manuscript.
- Structure of Sorafenib is missing in the manuscript. It may be added for readers convenience.
We thank referee for his/her remark. The structure of Sorafenib drug was added to Table 1 in the revised version of the manuscript.
- There are missing footnotes in the Table 2 (a, b, c, d, e)
We thank referee for this remark. The appropriate footnotes are added in the Table 2.
- Is Chang liver correct name or it should be clarified as Chang liver (HeLa)?
We agree with referee remark. Indeed, Chang liver of HeLa type was used in the biological activity studies. The appropriate correction has been made all along the revised version of the manuscript.
- It appears that not all of the synthesized substances are dry. Please note that the peak height ratio (I can't integrate from the picture) of water and DMSO varies from compound to compound. It looks like compound 12 contain much more water than compound 14.
All synthesized compounds were dried under vacuum after the working-up procedure. We assume that the presence of water molecules in NMR spectra is related with the low “dryness” of used DMSO-d6 solvent.
- Analytical data and spectra (in the SI) of compound 13 are missing
The analytical data of compound 13 are missing, because the synthesis of this compound was earlier reported, and the recorded spectra were found to be identical. The corresponding reference is given in the manuscript (see lines 384-385).
- Russian letters on NMR spectra axis labels
We thank referee for his/her remark. The appropriate correction has been made in the revised version of the of the Supplementary Material file.
- It seems that solubility of compound 10 in DMSO-d6 it rather low, 1H NMR spectrum of compound 10 is quite noisy, 13C NMR spectrum is even worse and S/N ratio is barely acceptable. Is it possible to find a more suitable deuterated solvent? Or acquire a spectrum at a higher temperature in DMSO, maybe the solubility will increase enough. The same about compound 15
Indeed, unfortunately the synthesized derivatives containing the nitro group are very poorly dissolved in DMSO-d6. However, as it was attested, the solubility in other solvents (CHCl3, MeOH, H2O, toluene, DMF) is even worse. We agree with referee comment that the intensities of the signals in the NMR spectra can be increased by performing NMR studies at higher temperature. But unfortunately, we don’t have easy access to high temperature NMR-spectroscopy at this moment.
- Also 1H NMR spectrum of compound 10 contains unidentified peak at 2.09 ppm. All NMR peaks should be labeled (water, dmso-d5/d6)
The unidentified peak in 1H NMR spectrum at 2.09 ppm belongs to acetone which was used for NMR tube washing. We apology that the trace amount of acetone still persist in the recorded spectra although the tube was well dried. All NMR peaks including the signals of water and DMSO-d6 molecules were labeled in the revised version of the Supporting Information file.